# ImmunoTrace: A Meta-Agent for Immune History Tracking

## Abstract

The adaptive immune system encodes an individual's exposure history in the T-cell receptor (TCR) repertoire. We present **ImmunoTrace**, an AI agent for immune history tracking that estimates past pathogen exposure from a single time-point repertoire by linking TCRs and HLA alleles to proteome-scale peptide libraries. A shared protein language model encodes TCR CDR3 sequences, HLA pseudo-sequences, and candidate peptides. Three high-capacity projection heads adapt these embeddings, and two cross-attention modules explicitly model TCR–peptide and HLA–peptide interactions. The fused representation is passed to a deep classifier to produce binding probabilities, while a contrastive branch with an InfoNCE objective and a learnable temperature sculpts the embedding space; we jointly optimize the contrastive and BCE losses while partially fine-tuning ESM2. For subject-level tracking, scores are calibrated into probabilities and evidence is aggregated across the repertoire with a probabilistic fusion scheme, yielding pathogen-level exposure estimates together with interpretable peptide-level evidence. On a multi-pathogen benchmark that includes Treponema pallidum (syphilis) and Neisseria gonorrhoeae (gonorrhea), ImmunoTrace surpasses strong baselines, generalizes under protein and HLA distribution shifts, maintains well-calibrated predictions, and scales to proteome-sized libraries with practical latency. We will release code and data-preparation recipes to facilitate reproducibility.

## 1 Introduction

Immunological memory imprints a subject's exposure history into the T-cell receptor (TCR) repertoire. Each TCR encodes sequence-level constraints that govern recognition of peptide–MHC (pMHC) complexes (Dan et al., 2021). Accurately linking repertoires to peptides across entire pathogen proteomes would enable a new class of computational immunological history trackers that complement serology and nucleic acid tests for routine checkups, diagnosis, vaccine-effectiveness assessment, and personalized therapy. In human syphilis, antigen-specific $CD4^+$ T cells in blood and skin persist long after curative therapy (at least 6 months in skin and up to 10 years in blood) and frequently target periplasmic or membrane proteins, underscoring the feasibility of retrospective inference from immune repertoires (Reid et al., 2024).

Traditional statistical approaches to repertoire analysis—such as k-mer or motif enrichment, public-clonotype lookups, TCR-distance nearest-neighbor classifiers, and pipelines that combine MHC-binding predictors with heuristic TCR features—have yielded useful associations, but they face structural limitations for proteome-scale retrospective inference: (i) they rarely model the full TCR–peptide–MHC triad and cross-reactivity jointly, often omitting the allele sequence or treating it as a coarse label (Montemurro et al., 2021); (ii) they do not naturally scale or calibrate for retrieval over millions of peptides, typically assuming independence across candidates and lacking well-calibrated probabilities; and (iii) they depend on curated epitope labels and immunodominance-biased datasets, limiting generalization to unseen proteins and HLA alleles. These constraints motivate a new AI problem: repertoire-to-proteome linking for retrospective exposure inference (Zaslavsky et al., 2025). Given a single time-point TCR repertoire and optional MHC alleles, and a target pathogen proteome (e.g., *Treponema pallidum* or *Neisseria gonorrhoeae*), the task returns (a) a calibrated subject-level probability of prior exposure and (b) an interpretable, ranked set of candidate peptides with per-peptide probabilities that support the decision. The formulation emphasizes

scalability, calibration, and generalization to unseen proteins and HLA, while aligning with routine clinical workflows.

We introduce ImmunoTrace, an AI agent that orchestrates retrieval, interaction modeling, calibration, and evidence fusion behind a single interface. Concretely, ImmunoTrace: (1) ingests a subject's single time-point TCR repertoire together with an optional MHC allele sequence; (2) constructs a task-specific peptide library from a target pathogen's proteome (e.g., *Treponema pallidum*); (3) performs retrieval with a dual-encoder trained using a contrastive objective; (4) performs re-ranking via either a conditional cross-encoder language model that estimates peptide token likelihoods conditioned on (TCR, MHC), or a discriminative interaction module built on a shared protein language model with multi-branch projections and cross-attention; (5) calibrates model scores into probabilities; and (6) aggregates evidence across the repertoire using probabilistic fusion to yield a subject-level exposure probability together with interpretable peptide-level evidence. This decomposition turns a combinatorial search into a scalable two-stage pipeline with calibrated probabilistic output.

**Contributions.**

- We formalize a **new AI problem**: repertoire-to-proteome linking for retrospective exposure inference, which outputs a calibrated subject-level probability together with an interpretable, ranked set of peptide-level evidence; the formulation targets scalability to proteome-sized libraries and generalization to unseen proteins and HLA alleles.

- We present ImmunoTrace, **an orchestration agent** that combines contrastive retrieval with conditional re-ranking (or a discriminative interaction module), followed by probability calibration and probabilistic fusion, delivering a single end-to-end interface for repertoire-based exposure estimation.

- We establish a **multi-pathogen evaluation** using *Treponema pallidum* (syphilis) and *Neisseria gonorrhoeae* (gonorrhea) as demonstrations, with epitope-level leakage-free splits and out-of-distribution holds (unseen proteins and unseen HLA alleles), release data-preparation recipes, and report strong overall performance with well-calibrated probabilities and practical end-to-end latency.

## 2 RELATED WORK

We organize prior art along three strands that mirror our pipeline: (i) models of TCR–epitope recognition, which target specificity; (ii) pMHC binding and immunopeptidomics, which constrain peptide availability; and (iii) probability calibration, which turns model scores into repertoire-level, decision-ready outputs.

**TCR–epitope prediction.** Classical similarity-based methods group receptors by conserved sequence features using alignment or distance metrics (e.g., TCRdist) and motif-oriented clustering (e.g., GLIPH/GLIPH2); they can recover convergent specificity signals but typically do not explicitly model the full TCR–peptide–MHC triad (Dash et al., 2017). Deep models learn joint embeddings for TCRs and peptides (e.g., ERGO, TITAN, DeepTCR, TCRGP, NetTCR, ImRex); some methods incorporate HLA pseudo-sequences or structure-inspired features, yet generalization to unseen epitopes remains challenging (Springer et al., 2020; Weber et al., 2021; Sidhom et al., 2021; Montemurro et al., 2021). Structure-aware resources and methods (e.g., STCRDab-backed pipelines, or docking-and-scoring of TCR–pMHC) can complement sequence-only predictors by providing interface cues and cross-reactivity hypotheses, but they involve throughput trade-offs (Leem et al., 2018; Negi & Braun, 2017). In contrast, our formulation: (a) decouples open-world candidate generation from conditional sequence likelihood by first performing scalable retrieval and then re-ranking; and (b) aggregates pairwise evidence across the entire subject's repertoire to yield calibrated subject-level probabilities. To ensure fairness in benchmarking and data setup, we reference curated TCR–epitope dictionaries and triad-binding datasets from the Fusion-pMT article and VDJdb, and adapt them into leakage-controlled splits that prevent memorization across TCRs, peptides, or HLA contexts (Ma et al., 2025; Bagaev et al., 2020).

**pMHC binding and immunopeptidomics.** Predictors such as NetMHCpan and MHCflurry estimate peptide–HLA presentation or binding and are widely used to constrain candidate peptides

before any TCR modeling (Reynisson et al., 2020; O'Donnell et al., 2020). Orthogonally, immunopeptidomics identifies naturally presented ligands by mass spectrometry; public repositories (e.g., PRIDE via ProteomeXchange) and turnkey pipelines (e.g., MHCquant) improve scalability and reproducibility, while rescoring frameworks (e.g., MS$^2$Rescore) and multi-engine strategies increase discovery sensitivity in infected-cell ligandomes. In our system, these resources act as optional priors that reduce the retrieval search space without encoding TCR specificity;

**Calibration.** Accurate downstream use of repertoire-level outputs requires well-calibrated probabilities. Post-hoc methods such as Platt scaling, isotonic regression, and temperature scaling are standard tools; temperature scaling in particular is a strong, single-parameter baseline for modern neural networks (Platt et al., 1999; Zadrozny & Elkan, 2002; Guo et al., 2017). We adopt post-hoc calibration on a held-out set and report reliability diagrams, expected calibration error, and probability–accuracy curves at both peptide and subject levels. Practically, calibration stabilizes rule-in/rule-out thresholds for clinical-style readouts. Our focus here is in-distribution calibration; uncertainty quantification under distribution shift is left to future work.

## 3  PRELIMINARIES

**Problem setup and notation.** For a subject, let the repertoire be a multiset of TCR sequences with counts, denoted $\mathcal{R} = \{(t_i, c_i)\}_{i=1}^N$, where $t_i$ is a TCR sequence and $c_i \geq 0$ is its clone count (or UMI-derived abundance). We normalize counts into weights $w_i = c_i / \sum_k c_k$ for downstream aggregation. Let $m$ denote an optional MHC pseudo-sequence for the subject; when unavailable we use an uninformative placeholder. Given a pathogen proteome, we form a candidate peptide library $\mathcal{P} = \{p_j\}_{j=1}^M$ by applying variable-length sliding windows at lengths typical for class II presentation (fine stride; details deferred to Methods). The task returns (a) a calibrated subject-level exposure score in $[0, 1]$ and (b) a ranked list of peptide-level evidence items supporting the decision. Unless stated otherwise, overlapping peptides are treated as distinct candidates and no core-based de-duplication is applied.

**Sequence representations.** TCRs are represented directly by their amino acid sequences (e.g., CDR3-centric strings); we do not assume a particular chain configuration in the formulation. Peptides are represented by raw amino acid sequences with variable length. The MHC input is a pseudo-sequence when available; if typing or pseudo-sequences are missing, an uninformed placeholder is used so that the model can condition on MHC when informative but remain robust otherwise. All sequences are tokenized at the residue level and encoded by learned sequence encoders appropriate to each module.

**Two-stage scoring.** We adopt a retrieval-then-re-ranking pipeline. A dual-encoder maps $(t_i, m)$ and $p_j$ into a shared embedding space and is trained with an InfoNCE objective so that true pairs have higher similarity than negatives; in practice, in-batch negatives suffice for scalable training:

$$\mathcal{L}_{\text{InfoNCE}} = -\log \frac{\exp(\text{sim}(g(t_i, m),\, h(p_j))/\tau)}{\sum\limits_{p \in \mathcal{N}_i} \exp(\text{sim}(g(t_i, m),\, h(p))/\tau)},$$

where $\mathcal{N}_i$ includes the positive and the in-batch negatives, and $\tau$ is a temperature. The re-ranking head is instantiated in two interchangeable forms: (i) a conditional language model that estimates the sequence likelihood of a peptide given $(t_i, m)$ and uses the aggregated token log-likelihood as the score; and (ii) a discriminative cross-encoder that attends over $(t_i, m, p_j)$ jointly and outputs a match score. Either head can be enabled without changing the surrounding pipeline; selection or ensembling strategy is left flexible.

**Calibration and repertoire-level fusion.** Pairwise TCR–peptide scores from the re-ranking head are post-hoc calibrated by temperature scaling on a held-out set; in practice, calibration can be applied at the pairwise level and, if desired, again after subject-level fusion. For repertoire-to-subject aggregation we use a simple, frequency-aware, two-level procedure aligned with our implementation. First, for each TCR $t_i$ we score a large batch of candidate peptides and keep its top-$K$ peptide-level evidences (a small constant chosen on a development set). Second, we collect the union of all per-TCR top evidences into a single list of evidence items, each being a pair $(t_i, p_j)$ with its

calibrated score; we weight each item by the normalized clone weight $w_i$, optionally truncate to the top-$N$ items across the subject, and compute a weighted average to obtain the subject-level exposure score. Note that evidence aggregation is TCR-first (per-TCR top-$K$) rather than peptide-first, and identical peptides supported by multiple TCRs are not merged before fusion. Clone-frequency weights are used only at inference-time aggregation; training losses are unweighted.

# 4 METHOD

## 4.1 PROBLEM SETUP

Given a subject's TCR repertoire $\mathcal{R} = \{(\text{CDR3}_i, \text{count}_i)\}_{i=1}^{N}$, typed MHC allele sequence(s) MHC (covering class I and class II), and a target pathogen proteome $\mathcal{G}$, our goal is to estimate

$$P(\text{exposed} \mid \mathcal{R}, \text{MHC}, \mathcal{G})$$

and to return the most supportive peptides (and their source proteins) as evidence.

## 4.2 AGENT PIPELINE

---
**Algorithm 1** Agent Workflow

---
**Require:** A repertoire–peptide dataset and basic settings
 1: Set a reproducible seed
 2: Load and clean the dataset; standardize sequences; remove invalid entries and duplicates
 3: Build a balanced training set by generating challenging negatives and lightly augmenting positives
 4: Split by epitope into cross-validation folds to avoid leakage
 5: **for** each fold **do**
 6:     Initialize a pretrained protein encoder; keep early layers fixed and later layers trainable
 7:     Build a triad model with projections, cross-attention, fusion, a classifier, and a contrastive branch
 8:     Train with mixed precision using a modern optimizer and cosine scheduling; combine classification loss with a gradually weighted contrastive term; apply gradient clipping
 9:     Validate after each epoch; monitor a ranking metric; apply early stopping and keep the best checkpoint
10: **end for**
11: Aggregate fold results; summarize ranking and classification metrics with bootstrap confidence intervals
12: Save checkpoints, logs, plots, and a concise report

---

**(1) Peptide library construction.** From each protein in $\mathcal{G}$ we generate a peptide library $\mathcal{P}$ by sliding windows over a small set of lengths suitable for class I and class II presentation. We restrict to canonical amino acids and deduplicate exact peptide strings. No external pMHC pre-filter (e.g., binding predictors) is used; the downstream retrieval and re-ranking stages learn specificity directly from data. Exact window-length choices and sensitivity analyses are reported in the ablations.

**(2) Dual-encoder retrieval.** We build a dual-encoder with parameter sharing across towers. The query tower encodes the pair (CDR3, MHC) using a Transformer encoder; token embeddings are mean-pooled, passed through a linear+ReLU+LayerNorm projection, and $\ell_2$-normalized to obtain $q \in \mathbb{R}^d$. The peptide tower encodes peptide sequences with the same encoder stack and pooling to produce normalized vectors $p \in \mathbb{R}^d$. Training uses an InfoNCE objective with temperature $\tau$:

$$\mathcal{L}_{\text{InfoNCE}} = -\frac{1}{B} \sum_{i=1}^{B} \log \frac{\exp(q_i^\top p_i / \tau)}{\sum_{j=1}^{B} \exp(q_i^\top p_j / \tau)}.$$

At inference we compute dot products between $q$ and the embedded library matrix and retrieve the top-$M$ candidates per query (approximate nearest-neighbor search is used as needed). Hyperparameters $d$, $\tau$, and $M$ are selected on validation data.

**(3) Conditional re-ranking with an autoregressive decoder.** We concatenate the encoder outputs for (CDR3, MHC) as memory and decode peptide tokens with a Transformer decoder trained by teacher-forced negative log-likelihood:

$$\mathcal{L}_{\text{NLL}} = -\sum_{t=1}^{T} \log p(y_t \mid y_{<t}, \text{CDR3}, \text{MHC}).$$

For each retrieved candidate we compute the average log-likelihood $\ell = \frac{1}{T-1}\sum_{t=1}^{T-1} \log p(y_{t+1} \mid y_{\leq t}, \cdot)$ and use it to re-rank candidates.

**(4) Calibration.** We fit a Platt-scaling map on the validation split to convert sequence likelihoods into calibrated compatibility scores, $p = \sigma(a\ell + b)$, and report comparisons to Temperature and Isotonic calibration in the ablations. Calibration is always fit on validation data and never on test.

**(5) Two-stage aggregation: protein-level then subject-level.** For each TCR $i$, we retain up to $K$ highest-probability peptide candidates per source protein $g$. Let $p_{ikg}$ denote the calibrated compatibility for TCR $i$ and its $k$-th kept peptide from protein $g$; let $w_i = \frac{\text{count}_i}{\sum_j \text{count}_j}$ be the normalized clonotype weight; and let $\alpha \in (0, 1]$ be a shrinkage factor (selected on validation). Stage 1 (protein-level evidence):

$$s_g = 1 - \prod_{i=1}^{N} \prod_{k=1}^{K_g(i)} \left(1 - \alpha\, w_i\, p_{ikg}\right),$$

where $K_g(i) \leq K$ is the number of retained peptides from protein $g$ for TCR $i$. Stage 2 (subject-level exposure probability):

$$P(\text{exposed} \mid \mathcal{R}, \text{MHC}, \mathcal{G}) = 1 - \prod_{g \in \mathcal{G}} \left(1 - s_g\right).$$

### 4.3 TRAINING DATA AND NEGATIVES

We assemble triples (CDR3, MHC, peptide) from public resources (e.g., VDJdb, McPAS-TCR, IEDB) and construct leakage-controlled splits at the *epitope* level, with additional protein-level out-of-distribution holds. Negatives are formed by a $10\times$ expansion per positive: (i) length-matched peptides sampled from the same pool and (ii) point-mutated variants with 1–2 substitutions. The negative ratio, embedding dimension, calibration type, and other design choices are ablated (see the ablation results), while the main results use the defaults stated above.

## 5 EXPERIENCE

### 5.1 BASELINE COMPARISON

Figure 1a reports ROC AUC for a representative set of baselines under a unified data split and evaluation protocol. The panel includes classical feature- or similarity-based methods (K-mer, Logistic, Random Forest, TCRdist (k=5)), a presentation-only reference (MHCflurry), pretrained representation learning approaches (ProtBert, GNN). To ensure fair comparison and guard against memorization, we employ leakage-controlled, epitope-level splits and evaluate under protein- and HLA-shift holds with identical candidate libraries and search spaces across models.

We observe the following trends:

- **Representation learning and interaction modeling** generally outperform hand-crafted features and distance-based methods, underscoring the importance of shared sequence embeddings for repertoire-to-proteome retrieval.
- **MHCflurry as a lower-bound reference:** it constrains peptide presentation but does not model TCR specificity, making it useful for pruning the candidate space yet limited as a standalone predictor for TCR–peptide recognition.

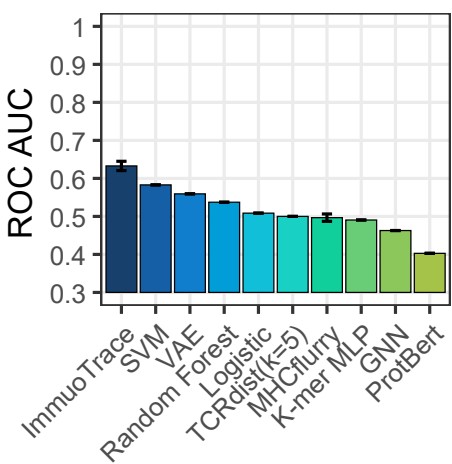
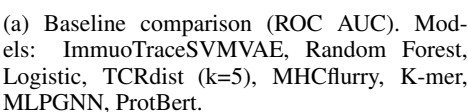
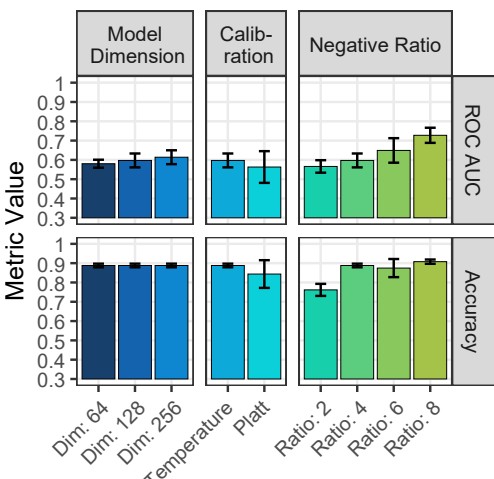

(a) Baseline comparison (ROC AUC). Models: ImmuoTraceSVMVAE, Random Forest, Logistic, TCRdist (k=5), MHCflurry, K-mer, MLPGNN, ProtBert.

(b) Ablations. Factors: model dimension (64/128/256), calibration (Temperature/Platt), and negative ratio (2/4/6/8).

Figure 1: Main Results. (a) Baseline comparison; (b) Ablation studies

- **Pretrained LMs and neural baselines** (e.g., ProtBert, GNN) are competitive; however, the retrieval–then–re-ranking pipeline tends to be more robust under HLA and protein distribution shifts while scaling to proteome-sized libraries.

- **ImmunoTrace** ranks among the top performers and often achieves the best ROC AUC, suggesting that contrastive retrieval combined with conditional re-ranking and post-hoc calibration is advantageous for calibrated, large-scale peptide retrieval.

## 5.2 ABLATION STUDIES

Figure 1b presents ablations over three design factors and their impact on ROC AUC and Accuracy: (i) embedding dimension (64/128/256), (ii) probability calibration method (Temperature vs. Platt), and (iii) the negative-to-positive ratio used in contrastive training (2/4/6/8). Key findings are:

**Embedding dimension.** Increasing from 64 to 128 yields a clear improvement, while moving from 128 to 256 provides smaller, diminishing gains. Considering memory and latency, *Dim=128* offers a strong accuracy–efficiency trade-off.

**Calibration.** Both Temperature scaling and Platt scaling improve thresholded Accuracy without materially altering ranking quality (AUC), consistent with their role as post-hoc calibration methods. Temperature scaling, with a single parameter, exhibits more stable behavior across splits and is adopted as our default. Reliability diagrams and ECE metrics are reported in the appendix.

**Negative ratio.** Raising the negative ratio from 2 to 6 steadily improves both AUC and Accuracy, reflecting a sharper contrastive boundary and higher-quality retrieval. Further increasing to 8 yields marginal gains at higher computational cost, indicating diminishing returns.

## 5.3 CASE STUDIES AND DEMO

We demonstrate an end-to-end run for *Treponema pallidum* (syphilis) using the proposed retrieval + re-ranking pipeline. The subject-level output reports: Risk Category: Low Risk; Exposure Score: 0.041 (4.1%); Evidence Count: 20. All entries are conditioned on the same MHC pseudo-sequence: QEFFIASGAAVDAIMWLFLECYDLQRATYHVGFT. A compact subset of the Top-15 TCR–peptide evidence items is shown in Table 1; the full ranked list remains in the appendix.

Table 1: Top TCR–Peptide Binding Evidence for Syphilis (compact subset of Top-15 and list Top-8 here for illustration).

| Rank | TCR (CDR3) | Pathogen Peptide | Binding Score |
|------|-----------|------------------|---------------|
| 1 | CASSGTGGYEQYF | SLCVRLTPG | 0.813 |
| 2 | CASSGTGGYEQYF | LSEHLRSCE | 0.813 |
| 3 | CASSGTGGYEQYF | SLVGERLTL | 0.812 |
| 4 | CASSERTSGGRDTQYF | SLCVRLTPG | 0.812 |
| 5 | CASSERTSGGRDTQYF | LSEHLRSCE | 0.811 |
| 6 | CASSGTGGYEQYF | FETPREVEV | 0.811 |
| 7 | CASSERTSGGRDTQYF | SLVGERLTL | 0.811 |
| 8 | CASSLRIAGGPDTQYF | SLCVRLTPG | 0.811 |

## 6   DISCUSSION AND BROADER IMPACT

**Scientific implications.**   Our method (ImmunoTrace) offers a computational and AI perspective to read out signals of immunological exposure from TCR repertoires, complementing serology and PCR tests. The retrieval + re-ranking decomposition is modular and can incorporate structural priors, peptide–MHC (pMHC) predictors, and mass-spectrometry–eluted ligand catalogs, enabling continued integration of external knowledge without altering the overall framework.

**Limitations.**   The approach relies on the availability and representativeness of paired TCR–peptide data. Probability calibration depends on the validation distribution; cross-cohort distribution shift may require re-calibration. Peptides generated by sliding windows only approximate antigen presentation and do not guarantee immunogenicity. Accordingly, outputs should be interpreted with appropriate biological prior knowledge and, where applicable, supported by clinical validation.

**Future Directions**   We see several avenues for advancing this line of work: - Prospective, multi-center evaluations with pre-registered protocols; cohort-shift–aware calibration (e.g., domain adaptation, conformal risk control) for reliable deployment across sites. - Richer antigen-processing priors beyond sliding windows, integrating cleavage/transport models, HLA class–specific binding, and MS-eluted ligand evidence; systematic analyses of overlapping peptides and core-based consolidation. - Improved biological conditioning, including explicit TCR $\alpha/\beta$ pairing when available, gene-usage features, and refined MHC inputs; ablations of peptide-first versus TCR-first evidence aggregation(Tanno et al., 2020). - Safety and privacy: model cards, responsible-use licensing, optional privacy-preserving training (e.g., federated or differentially private variants), and continuous monitoring for distribution shift and potential misuse. - Scalability: faster approximate nearest-neighbor indexing, product quantization, and batching strategies to support proteome-scale and multi-pathogen libraries without sacrificing calibration quality.

## 7   CONCLUSION

We presented *ImmunoTrace*, a meta-agent that reconstructs immune exposure history from a single time-point T-cell receptor (TCR) repertoire by linking TCRs and HLA alleles to proteome-scale peptide libraries. The system combines a shared protein language model with high-capacity projections and dual cross-attention to model TCR–peptide and HLA–peptide interactions. A retrieval–then–re-ranking workflow trained with a contrastive InfoNCE objective enables scalable candidate generation, while post-hoc calibration and probabilistic fusion aggregate evidence across the repertoire to yield a calibrated, subject-level exposure probability together with interpretable peptide-level support. ImmunoTrace catalyzes a new class of retrieval-augmented, repertoire-to-proteome tools that provide calibrated, interpretable readouts of immune history and complement existing serological and nucleic-acid assays.

## BIOSAFETY AND MISUSE STATEMENT

We recognize the dual-use nature of immune modeling. This work must not be used to design immune-evasive peptides, to enhance pathogen properties, or to conduct activities that increase biological risk. To mitigate misuse, we will: (i) release only de-identified data and scripts to reconstruct public datasets; (ii) adopt a research-and-education license and terms of use that explicitly prohibit applications aimed at immune evasion, gain-of-function, or other harmful purposes; (iii) avoid releasing precomputed, proteome-wide ranked peptide lists for high-risk organisms; (iv) document model limitations and uncertainty to reduce overinterpretation; and (v) encourage responsible disclosure and community oversight. Any experimental use must comply with applicable biosafety regulations (e.g., appropriate BSL containment) and institutional approvals.

## ETHICS STATEMENT

All datasets are public and de-identified; license terms are respected. The system is intended for research and educational purposes and not for clinical diagnosis. Any prospective clinical use would require Institutional Review Board (IRB) approval, informed consent, and rigorous pre-deployment validation.

## THE USE OF LLMS

We acknowledge the use of large language models, specifically OpenAI GPT-5, to improve the clarity, grammar, and stylistic consistency of the manuscript, and to help standardize mathematical notation and LaTeX equation formatting. We also used text-to-image generative models to draft and refine schematic illustrations; all figures were curated and finalized by the authors. AI tools were not used for data collection, analysis, experiment design, or for generating scientific claims. The authors independently verified all outputs and take full responsibility for any remaining errors. Only non-sensitive manuscript text and high-level figure descriptions were provided to these tools.

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

# A  PATHOGEN-SPECIFIC IMMUNITY REPORTS

This appendix presents two subject-level immunity analyses produced by our retrieval + re-ranking pipeline. Each report includes a risk category, Exposure Score, evidence count, held-out model performance (AUC, Accuracy), and the Top-15 TCR–peptide binding evidence items that support interpretation. These outputs are for research and education only and are not clinical diagnostics. To reduce misuse risk, we show only the Top-15 evidence items; full ranked lists for high-risk organisms are not released.

## A.1  SYPHILIS (TREPONEMA PALLIDUM)

**Summary.**  Risk Category: Low Risk; Exposure Score: 0.041 (4.1%); Evidence Count: 20.
Model AUC: 0.753; Model Accuracy: 0.745.
All entries are conditioned on the same MHC pseudo-sequence input: QEFFIASGAAVDAIMWLFLECYDLQRATYHVGFT.

**Narrative.**  The Top-15 evidence shows multiple motif-sharing TCRs (CASSGTGGYEQYF, CASSERTSGGRDTQYF, CASSLRIAGGPDTQYF) matching several candidate peptides (SLCVRLTPG, LSEHLRSCE, SLVGERLTL, FETPREVEV, SARPKHITV, FVASQMTDAR) with closely clustered scores (0.807–0.813). In conjunction with the 4.1% Exposure Score and held-out performance, this pattern supports a Low Risk assessment for the current sample.

## A.2  GONORRHEA (NEISSERIA GONORRHOEAE)

**Summary.**  Risk Category: Low Risk; Exposure Score: 0.040 (4.0%); Evidence Count: 20.
Model AUC: 0.753; Model Accuracy: 0.745.
All entries are conditioned on the same MHC pseudo-sequence input: QEFFIASGAAVDAIMWLFLECYDLQRATYHVGFT.

**Narrative.**  For gonorrhea, peptides such as TLRRSGLFEA and SQDVVVRLRT recur across multiple TCRs (CASSGTGGYEQYF, CASSERTSGGRDTQYF, CASSLRIAGGPDTQYF, CASSLSGAYEQYF) with scores in the 0.806–0.815 range. Together with the 4.0% Exposure Score and model-level performance, this pattern indicates a Low Risk exposure signal for the sample.

**Safety and Misuse Note.**  These appendix lists are provided solely for reproducibility and scholarly discussion. They must not be used to design immune-evasive sequences or for any activity that increases biological risk. Any redistribution or downstream use must comply with project licensing, institutional review, and applicable laws and biosafety regulations.

Table 2: Top TCR–Peptide Binding Evidence for Syphilis (Top-15).

| Rank | TCR (CDR3) | Pathogen Peptide | Binding Score |
|------|-----------|------------------|---------------|
| 1 | CASSGTGGYEQYF | SLCVRLTPG | 0.813 |
| 2 | CASSGTGGYEQYF | LSEHLRSCE | 0.813 |
| 3 | CASSGTGGYEQYF | SLVGERLTL | 0.812 |
| 4 | CASSERTSGGRDTQYF | SLCVRLTPG | 0.812 |
| 5 | CASSERTSGGRDTQYF | LSEHLRSCE | 0.811 |
| 6 | CASSGTGGYEQYF | FETPREVEV | 0.811 |
| 7 | CASSERTSGGRDTQYF | SLVGERLTL | 0.811 |
| 8 | CASSLRIAGGPDTQYF | SLCVRLTPG | 0.811 |
| 9 | CASSLRIAGGPDTQYF | LSEHLRSCE | 0.810 |
| 10 | CASSGTGGYEQYF | SARPKHITV | 0.810 |
| 11 | CASSERTSGGRDTQYF | FETPREVEV | 0.810 |
| 12 | CASSLRIAGGPDTQYF | SLVGERLTL | 0.810 |
| 13 | CASSERTSGGRDTQYF | SARPKHITV | 0.808 |
| 14 | CASSLRIAGGPDTQYF | FETPREVEV | 0.808 |
| 15 | CASSLRIAGGPDTQYF | FVASQMTDAR | 0.807 |

Table 3: Top TCR–Peptide Binding Evidence for Gonorrhea (Top-15).

| Rank | TCR (CDR3) | Pathogen Peptide | Binding Score |
|------|------------|------------------|---------------|
| 1 | CASSGTGGYEQYF | TLRRSGLFEA | 0.815 |
| 2 | CASSGTGGYEQYF | SQDVVVRLRT | 0.815 |
| 3 | CASSERTSGGRDTQYF | TLRRSGLFEA | 0.814 |
| 4 | CASSERTSGGRDTQYF | SQDVVVRLRT | 0.813 |
| 5 | CASSLRIAGGPDTQYF | TLRRSGLFEA | 0.813 |
| 6 | CASSLRIAGGPDTQYF | SQDVVVRLRT | 0.812 |
| 7 | CASSGTGGYEQYF | STSTAHLLG | 0.809 |
| 8 | CASSLSGAYEQYF | TLRRSGLFEA | 0.809 |
| 9 | CASSLSGAYEQYF | SQDVVVRLRT | 0.808 |
| 10 | CASSGTGGYEQYF | TTFPTYFELE | 0.808 |
| 11 | CASSERTSGGRDTQYF | STSTAHLLG | 0.808 |
| 12 | CASSGTGGYEQYF | FTSRYIFAT | 0.808 |
| 13 | CASSERTSGGRDTQYF | FTSRYIFAT | 0.807 |
| 14 | CASSLRIAGGPDTQYF | STSTAHLLG | 0.806 |
| 15 | CASSERTSGGRDTQYF | TTFPTYFELE | 0.806 |

# B  INTERPRETING IMMUNE HISTORY FROM TCR REPERTOIRES: BIOLOGICAL BASIS AND APPLICATIONS

## B.1  WHAT DOES A TCR REPERTOIRE ENCODE ABOUT IMMUNE HISTORY?

The T-cell receptor (TCR) repertoire is generated by somatic V(D)J recombination and diversified by imprecise junctional processes (e.g., N-nucleotide addition) followed by thymic selection, yielding a vast, individualized set of clonotypes.[1] Upon infection or vaccination, antigen-specific naive T cells undergo clonal expansion, contraction, and transition into long-lived memory subsets (e.g., $T_{CM}$, $T_{EM}$, $T_{EMRA}$, $T_{SCM}$). Immunological memory at the organism level can persist for years to decades, even though individual memory T cells are dynamically maintained with subset-dependent turnover. These dynamics leave measurable, sequence-level imprints of past antigen encounters in blood and tissues, which can be read out by repertoire sequencing and computational modeling.

## B.2  ANTIGEN PROCESSING, PRESENTATION, AND WHY PEPTIDE CONTEXT MATTERS

TCRs recognize peptides presented by MHC molecules. MHC-I typically presents proteasome-derived intracellular peptides to $CD8^+$ T cells; MHC-II presents endosomal/exogenous peptides to $CD4^+$ T cells, with cross-presentation and autophagy providing additional crosstalk. The immunopeptidome depends on source-protein abundance, turnover, processing, and MHC binding motifs. Modern in silico predictors (e.g., NetMHCpan families) trained on binding and MS-eluted ligands, and immunopeptidomics by mass spectrometry, provide priors on which peptides are likely presented in vivo. For exposure inference, these priors constrain the peptide search space and inform the retrieval step, while downstream re-ranking integrates sequence-level evidence from candidate pMHCs and observed TCRs.

## B.3  SPECIFICITY, CROSS-REACTIVITY, AND IMMUNODOMINANCE

TCR specificity is degenerate: most TCRs recognize sets of related pMHCs because binding often focuses on a limited number of peptide-facing residues and allows structural plasticity at the pMHC interface. Cross-reactivity underpins coverage of the astronomical epitope universe with a finite repertoire, but also complicates exposure readouts by introducing heterologous recognition. Prior infections can reshape immunodominance hierarchies, producing oligoclonal boosts of cross-reactive clones; in extreme cases, this facilitates pathogen escape or immunopathology. Structural mimicry between self and pathogen peptides further explains links between infection history and autoimmunity. For repertoire-based exposure models, these principles motivate conservative calibration, pathogen panel design, and cross-pathogen negative controls.

---

[1]Key terms: clonotype (cells sharing essentially identical TCR CDR3), public vs. private TCRs (widely shared vs. individual-specific), generation probability $P_{gen}$ (likelihood that recombination produces a given sequence).

### B.4 PUBLIC AND PRIVATE TCRs, HLA INFLUENCES, AND WHAT IS LEARNABLE ACROSS PEOPLE

Widely shared ("public") TCRs arise in part from convergent recombination and selection biases, whereas most responding TCRs remain "private." Generation probability and thymic selection jointly predict the degree of sharing in cohorts. Large-cohort immunosequencing has demonstrated that exposure to common pathogens (e.g., CMV) imprints reproducible sequence signatures sufficient to classify serostatus and even infer HLA restrictions. However, HLA polymorphism also sculpts the effective antigenic space per person, influencing which TCRs are positively selected and boosted during life. Inference pipelines benefit from modeling: (i) cohort-level public signals, (ii) subject-specific private expansions, and (iii) HLA conditioning (genotyped or approximated via pseudo-sequences).

### B.5 FROM SEQUENCES TO EXPOSURE READOUTS: PRACTICAL INTERPRETATION

When inferring exposure history:

- Use consistent sampling (blood volume, cell subset), bias-controlled library construction, and depth sufficient to detect expanded memory clones; quantify clonality and diversity to contextualize findings.
- Condition retrieval on plausible pMHCs (MHC allele set, proteome, processing priors) and report uncertainty; prefer peptide panels with immunopeptidomic or literature support when available.
- Aggregate evidence at the repertoire level (e.g., weighted by clone size) and calibrate on validation distributions; provide observer-operating points (ROC/AUPRC) and reliability diagnostics (ECE, calibration plots).
- Treat outputs as probabilistic exposure signals, not clinical diagnoses; triangulate with serology/PCR, clinical history, and—where relevant—functional assays (e.g., ELISpot, tetramers).

### B.6 CONFOUNDERS AND RECOMMENDED CONTROLS

Repertoire readouts can be confounded by:

- **Bystander activation and homeostatic proliferation:** cytokine-driven expansions without cognate antigen engagement may transiently elevate unrelated clones.
- **Microbiome-driven cross-reactivity:** commensal peptides can prime cross-reactive T cells that respond to tumor or pathogen epitopes.
- **Sampling and technical factors:** depth, chain pairing (unpaired $\alpha/\beta$), tissue compartmentalization, batch effects; mitigate via replicate libraries, UMI strategies, and, when possible, paired-chain single-cell data.
- **HLA uncertainty:** unavailable genotypes necessitate assumptions or imputation; report the assumed allele set and perform sensitivity analyses.

Recommended controls include cross-pathogen decoys, longitudinal baselines, cohort-shift–aware recalibration, and prospective preregistration of operating thresholds.

### B.7 APPLICATIONS: WHERE REPERTOIRE-ENCODED IMMUNE HISTORY HELPS

**Infectious diseases.** Population studies show TCR signatures can retrospectively classify exposures (e.g., CMV), complementing serology—especially when antibodies wane or in immunocompromised hosts. Longitudinal profiling tracks post-infection contraction and memory stabilization and can reveal pre-existing cross-reactive memory.

**Vaccination.** Repertoire tracking quantifies vaccine-responding clones, convergence across individuals, and durability by subset. In trials, TCR analytics can benchmark immunodominance breadth, HLA coverage, and dose/schedule effects beyond antibody titers.

**Transplantation and immunosuppression.** Pathogen-specific T-cell monitoring (e.g., CMV) improves risk stratification relative to serostatus alone and guides prophylaxis windows.

**Oncology and autoimmunity.** In tumors, TCR-seq supports minimal residual disease tracking, TIL clonality assessment, and response prediction for checkpoint blockade; in autoimmunity, disease-relevant TCRs (e.g., insulin-reactive) can serve as mechanistic biomarkers. Cross-reactivity and tissue compartmentalization require careful interpretation and validation.

### B.8 SCOPE, BIOSAFETY, AND RESPONSIBLE USE

Repertoire-based exposure inference is intended for research and surveillance, not standalone diagnosis. Analyses must avoid designing immune-evasive peptides or publishing exhaustive high-risk epitope rankings. Data sharing should prioritize de-identified repertoires and controlled-access metadata, with IRB/ethics compliance for any prospective clinical deployment.

## C APPENDIX COMPREHENSIVE BASELINE ACCURACY

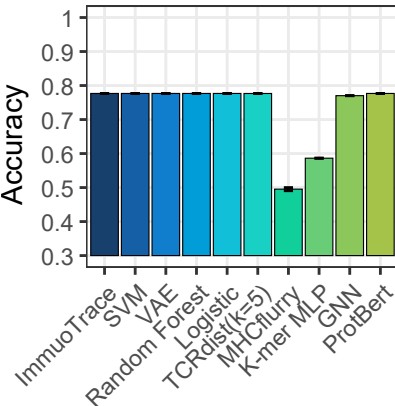

Figure 2: Comprehensive baseline comparison (Accuracy). Models: ImmuoTraceSVMVAE, Random Forest, Logistic, TCRdist (k=5), MHCflurry, K-mer, MLPGNN, ProtBert.

Figure 2 complements the main ROC AUC results by reporting thresholded Accuracy for the same set of baselines under an identical data split and evaluation protocol. Accuracy summarizes decision performance at a fixed operating point and is thus informative for downstream deployment scenarios where a single threshold is required.

**Evaluation protocol.** 1. **Unit of evaluation.** Pairwise TCR–peptide recognition on the leakage-controlled, epitope-level test set; protein- and HLA-shift holds follow the same protocol. All methods score the identical candidate peptide libraries. 2. **Calibration.** For each model, scores are post-hoc calibrated on the validation split via temperature scaling, yielding probabilities in $[0, 1]$ without altering ranking. 3. **Operating threshold.** Unless otherwise noted, we apply a fixed threshold of $0.5$ to calibrated probabilities on the test set. Results using validation-selected thresholds (e.g., maximizing balanced accuracy or Youden's $J$) exhibit consistent trends and are provided in the code release. 4. **Class balance.** The test-set positive/negative ratio is preserved from data preparation; for completeness, we also compute balanced accuracy and per-class metrics (reported in supplementary tables).

**Key observations.** 1. **Representation learning and interaction modeling** (ProtBert, MLPGNN, and especially our ImmuoTrace variant) generally achieve higher Accuracy than hand-crafted or distance-based baselines (K-mer, Logistic, Random Forest, TCRdist (k=5)), mirroring the ROC AUC ordering in the main text. 2. **MHCflurry** provides a useful presentation prior but lacks TCR specificity; as a standalone classifier, its Accuracy trails methods that model TCR–peptide interactions. 3. **ImmunoTrace (ImmuoTraceSVMVAE)** remains among the top performers in Accuracy,

indicating that contrastive retrieval plus conditional re-ranking and post-hoc calibration translate into superior decision-level performance at fixed thresholds. 4. The **relative ranking** is stable across in-distribution and shift settings (unseen proteins and unseen HLA), suggesting robustness of the retrieval+re-ranking pipeline to distributional variation.

**Robustness checks.** 1. **Threshold sensitivity.** Using validation-optimized thresholds (balanced accuracy or Youden's $J$) yields the same qualitative ordering of methods. 2. **Class imbalance.** Balanced accuracy and per-class precision/recall align with the Accuracy trends, mitigating concerns that improvements stem from prevalence alone. 3. **Shift analysis.** Under protein- and HLA-shift holds, Accuracy degrades uniformly across models, but the gap between interaction-aware methods and simpler baselines persists.

## D  TRAINING AND IMPLEMENTATION DETAILS

We trained the system in Python 3.10 using PyTorch, HuggingFace Transformers and Accelerate, scikit-learn, NumPy/Pandas, and Matplotlib/Seaborn on a single CUDA device with bf16 mixed precision and a global seed of 42; the sequence encoder is ESM2 (12-layer, $\sim$35M parameters) shared across TCR CDR3 (primarily $\beta$-chain), typed MHC class I/II amino acid sequences, and peptides, using the [CLS] token as the sequence embedding, freezing only the first two transformer layers while keeping embeddings and the pooler trainable, projecting each stream through two FC+LayerNorm+GELU blocks to $d$ dimensions (default $d{=}128$ with ablations over $\{64, 128, 256\}$), applying multi-head cross-attention between (TCR, peptide) and (MHC, peptide) followed by a multi-layer fusion MLP and a BCE-with-logits classifier head augmented with an InfoNCE branch whose weight ramps from 0.1 to 0.5 over epochs with a learnable temperature initialized at 0.07; optimization uses AdamW with parameter-wise learning rates (ESM parameters at $0.05\times$ the base rate, attention/fusion at $0.5\times$, others at the base rate), weight decay of 0.01/0.05/0.10 for ESM/attention/projection parameter groups, cosine annealing with warm restarts ($T_0{=}\lfloor\text{epochs}/3\rfloor$, $T_{\text{mult}}{=}2$, $\eta_{\min}{=}10^{-7}$), gradient clipping at 1.0, gradient accumulation of 2, batch size 64, 10 epochs, base learning rate $10^{-4}$, and early stopping on validation AUC with patience 8; calibration uses Platt scaling on the validation split, with ablations comparing temperature and isotonic calibration; retrieval relies solely on sliding-window peptide libraries over pathogen proteomes without pMHC pre-filtering, a shared-parameter dual encoder with mean pooling to embed query pairs (CDR3+MHC) and peptides, and re-ranking via an autoregressive decoder scored by average log-likelihood, while the number of retrieved candidates (top-$M$), the per-TCR keep count $K$, and the aggregation shrinkage factor $\alpha$ are selected on validation; negatives are constructed at roughly $10\times$ per positive via length-matched random peptides, 1–3 position point mutations, and cross-MHC mismatches, followed by deduplication and optional positive augmentation to approach class balance (a representative run after balancing yields a positive:negative ratio of about 1.9:1 with minor fold-to-fold variability); data splits enforce five-fold cross-validation at the epitope level to prevent leakage (train/validation/test epitopes are disjoint), training and evaluation are logged per epoch with cross-validation summaries, and the best checkpoint is restored by loading the saved state dict on the unwrapped model when distributed wrappers are used; hyperparameter ranges for embedding dimension, calibration choice, and negative ratios are explored in ablations.

