# OpenReview forum: "ImmunoTrace: A Meta-Agent for Immune History Tracking"
_ICLR.cc/2026/Conference — ICLR 2026 Conference Withdrawn Submission_

### Official Review · Reviewer_3fx6 · 2025-10-17

**Soundness:** 1
**Presentation:** 1
**Contribution:** 3
**Rating:** 2
**Confidence:** 4

**Summary:**

This manuscript proposes a novel computational framework that integrates TCR, MHC, and peptide sequence information to infer an individual’s exposure to specific antigens, enabling the reconstruction of past pathogen exposure. The method involves several key stages, including peptide library construction, retrieval, conditional re-ranking, calibration, and aggregation at both the protein and subject levels. The authors demonstrate the utility of their approach by analyzing exposure to Treponema pallidum (syphilis) and Neisseria gonorrhoeae (gonorrhea). Additionally, the method shows improved performance on public benchmark datasets such as VDJdb, McPAS-TCR, and IEDB.

**Strengths:**

1. The study addresses an important problem, reconstructing immune exposure history, which has significant implications for immunotherapy design and disease diagnosis.
2. Unlike existing approaches that model TCR-epitope, peptide-MHC, and antigen processing separately, this work integrates these components into a unified framework for more robust immune modeling.
3. The use of a pre-trained protein language model (ESM2) as the backbone improves performance without requiring extensive retraining.
4. Instead of directly predicting exposure outcomes, the method provides interpretable evidence supporting immune history inference, facilitating biological interpretation and professional analysis.

**Weaknesses:**

1. Although the manuscript claims to introduce a novel approach for immune history inference, the presented experiments do not adequately support this claim. The two main evaluations, ROC-AUC analysis and case studies, are insufficient to validate immune history tracking.
2. The ROC-AUC experiments rely on datasets such as VDJdb, which are designed for TCR–pMHC binding prediction rather than immune history inference. To substantiate the main claim, the authors should include experiments evaluating whether the proposed method can detect pathogen exposure over varying time intervals, rather than focusing solely on binding prediction tasks.
3. The case studies lack justification and validation. The rationale for selecting Treponema pallidum and Neisseria gonorrhoeae is not clearly explained, and no analysis is provided to verify whether the model’s inferred evidence is biologically correct or indicative of true exposure.
4. The data preprocessing and input definition are unclear. The manuscript mentions the use of clone counts or UMI information, which are not available in datasets such as McPAS-TCR. Similarly, subject-level metadata are missing for many datasets and samples, yet the authors do not clarify how these data were obtained or handled.
5. The method description contains several undefined notations, duplicated definitions, and a confusing organizational structure. This section would benefit from reorganization and clearer notation definitions.
6. The background and introduction are incomplete: some referenced methods are not properly cited, and recent relevant works are not discussed, limiting the contextual grounding of the paper.

Please check questions section for detailed information.

**Questions:**

### **Soundness**
Overall, the experimental details are not clearly specified, and the current results do not sufficiently support the authors’ claim of tracking immune history.

1. **Line 260:** Which GNN architecture is used? The appendix mentions "MLPGNN". Please clarify what this refers to. If it denotes a GNN combined with an MLP, provide the detailed configuration of both components.
2. **Line 260:** ProtBERT is typically used for feature extraction only. How is the final prediction obtained from these features? Are the global or local features used?
3. **Line 259:** Why was $ k = 5 $ chosen for TCRdist?
4. **Line 292:** Models such as PISTE (Feng et al., 2024, Nat. Mach. Intell.) and TULIP (Meynard-Piganeau et al., 2024, PNAS) also use TCR, MHC, and peptide inputs. Why were these not included in the comparison?
5. **Line 255:**
   1. Is the evaluation dataset derived from public datasets (e.g., VDJdb) or specific datasets for Treponema pallidum and Neisseria gonorrhoeae?
   2. Is the reported ROC-AUC measuring TCR-pMHC binding or past pathogen exposure? If it represents exposure, what is the time interval between exposures?
   3. Please describe the evaluation benchmark used in detail.
6. **Line 305:** What is the embedding dimension mentioned here? Does it apply only to the customized modules or also to ESM2? Please also specify which version of ESM2 was used.
7. **Figure 1:** The accuracy appears high, but the ROC-AUC values are between 0.5 and 0.6 for most models. Given that there are 10× more negative samples, please also report recall and precision to determine whether the model performs disproportionately well on negative samples.
8. **Line 314:** What does the "negative ratio" refer to? Is it defined for the training dataset or for evaluation?
9. **Line 318:**
   1. Why were these two case studies chosen? Please provide a brief biological background for each antigen.
   2. Describe the key observations rather than just reporting numerical results. Also, explain the "report notes" — for instance, what does "Risk Category" represent?
   3. The output shows "low risk."" Is there any ground truth supporting this label, or is it purely model-based without external validation?
   4. The term "evidence" is unclear, while all listed peptides have high binding scores, so why does the final decision indicate "low risk"? Please clarify and discuss.
   5. What do the peptides in `Table 1` represent? Are they core regions for syphilis?
   6. Why are the gonorrhea results not shown or discussed here?

### **Presentation**
Overall, several background citations are missing, multiple notations are undefined, and the descriptions are often redundant or unclear. The manuscript would benefit from substantial reorganization and editing.

1. **Line 062:** The tuple "(TCR, MHC)" is not defined. Please clarify what this notation represents instead of using a tuple form.
2. **Line 093:** Both TCRdist and GLIPH are mentioned, but only TCRdist is cited.
3. **Line 094:** TCRGP and ImRex are listed but not cited.
4. **Line 094:** Recent transformer-based methods such as MixTCRpred (Croce et al., 2024, Nat. Commun.) and TULIP (Meynard-Piganeau et al., 2024, PNAS) should also be discussed, especially since the proposed approach is transformer-based (ESM2).
5. **Line 110:** PRIDE, ProteomeXchange, MHCquant, and MS²Rescore are mentioned but not cited.
6. **Lines 143–166:** These paragraphs describe methodological details and should be moved to the `Methods` section.
7. **Line 147:** Please define $ g $ and $ h $ in the equation, and specify how similarity (`sim`) is computed.
8. **Line 172:** The prior sections focus on TCRs in general, but here the text suddenly refers to CDR3 without introduction. Please specify whether CDR3 alpha, beta, or both are used.
9. **Line 180:** The pseudo-code block is largely uninformative and better suited for describing experimental setup (e.g., "set seed" and "save logs") rather than the method itself.
10. **Line 188:** The model workflow should be clearly described here, but the explanation is currently unclear.
11. **Line 201:** The sliding window size is not specified and should be reported or included in ablation studies but not found.
12. **Line 211:** $ B $ is undefined. Does it represent the batch size?
13. **Line 215:** $ M $ is used earlier in preliminaries (Line 130). Are they referring to the same quantity?
14. **Line 221:** $ T $ is not defined.
15. **Line 230:** The authors use "TCR $ i $" instead of CDR3, are these equivalent?
16. **Line 231:** The variable $ p_{ikg} $ is introduced, how does it differ from $ p $? Please clarify.
17. **Line 239:**
    1. How many samples were selected from each public dataset?
    2. How many samples contain MHC allele information? Please report counts for both class I and class II.
    3. How many unique subjects are included?
    4. For datasets such as McPAS-TCR, how were missing UMI/clone counts handled?

---

### Official Review · Reviewer_dSMh · 2025-10-27

**Soundness:** 1
**Presentation:** 1
**Contribution:** 1
**Rating:** 2
**Confidence:** 4

**Summary:**

The paper leverages contrastive learning and a protein language model (ESM-2) for subject-level inference of pathogenic exposure given a subject's TCR repertoire, MHC, and pathogen proteome. The proposed approach aggregates predictions from TCR-pMHC predictions to infer subject-level outcomes. Experimental results on TCRpMHC tasks show competitive AUROC compared to alternative TCRpMHC baselines.

**Strengths:**

- Subject-level inference of pathogenic exposure given a subject's TCR repertoire is an important problem.
- Experimental results on TCR-peptide tasks show competitive AUROC compared to alternative TCR-peptide baselines.

**Weaknesses:**

- The paper makes several claims without providing experimental or theoretical evidence, such as:
1) Generalization to unseen proteins and HLA alleles: No experimental results on unseen proteins or HLAs are provided.
2)  Well-calibrated probabilities: The paper does not provide calibration metrics.
3) Scalability: It's unclear how the proposed approach is scalable. There are an estimated  $10^{15}$ TCRs, $10^{5}$ HLAs, and  $20^{10}$ peptides. Given that the proposed approach relies on labeled CDR3, MHC, and peptide triplets, it will require a significant amount of data to generalize.
- The proposed approach is a straightforward combination of ESM-2 and contrastive learning with little technical novelty.
- Given the limited technical contributions, the experimental results are underwhelming:
1)  The paper benchmarks with TCR-peptide approaches but does not evaluate or discuss the proposed approach against competitive baselines for repertoire-level pathogenic exposure prediction, including references [1, 2, 3].
2) The writing could be improved to better emphasize the experimental setup and the motivations for the modeling approach relative to alternative baselines.

**References**
- [1] Chapfuwa et al.,  "Scalable Universal T-Cell Receptor Embeddings from Adaptive Immune Repertoires", ICLR 2025
- [2] Widrich et al., "Modern hopfield networks and attention for immune repertoire classification", NeurIPS 2020
- [3] Pradier et al., "Airiva: a deep generative model of adaptive immune repertoires",   MLHC 2023

**Questions:**

- How many CDR3, MHC, and peptide triplets are used for training? Also what is the distrubution in terms pMHC?
- Why are the V- and J-genes not used to encode the TCRs along with the CDR3?
- How does the proposed approach compare to alternative repertoire-level pathogenic inference approaches, including [1, 2, 3]?
- Could you provide benchmarks for repertoire-level tasks?
- Could you provide calibration results?
- Could you provide an ablation study, including ESM-2 embeddings and contrastive learning?
- How does the proposed approach scale in #TCRS, #HLAs and #peptides?
- How does the proposed approach account for homologous pathogens?

---

### Official Review · Reviewer_uWRB · 2025-10-30

**Soundness:** 2
**Presentation:** 2
**Contribution:** 1
**Rating:** 2
**Confidence:** 4

**Summary:**

The paper introduces ImmunoTrace, a framework for predicting an individual's prior pathogen exposure directly from their T-cell receptor repertoire. The idea is to jointly encode TCRs, HLA pseudo-sequences, and candidate peptides into a shared embedding space, using contrastive learning and cross-attention to capture potential binding relationships.
The model further applies post-hoc calibration and a probabilistic fusion step to summarize clone-level predictions into a subject-level exposure score. The authors frame this as a scalable “meta-agent” that can operate on proteome-scale peptide libraries and produce interpretable peptide-level evidence. Results on a small benchmark (syphilis and gonorrhea) show slightly higher ROC-AUC than several sequence-based baselines. The concept is interesting and potentially impactful for computational immunology, but in its current form, the work feels overstated relative to the experimental evidence.

**Strengths:**

Modeling immune exposure directly from repertoire data is an ambitious and worthwhile goal, especially as TCR sequencing grows in biomedical relevance. Framing it as a unified retrieval to re-ranking and fusion pipeline is conceptually neat and easy to follow. Many immunoinformatics models stop at classification scores, so the attempt to incorporate explicit calibration is a good step forward. Even if the results are limited, this shows awareness of reliability and uncertainty, which is rare in this area. Building on ESM2 embeddings is sensible and technically solid. Using shared language model features across TCR, peptide, and HLA tokens reflects a good understanding of how to leverage recent foundation models for biological sequences.

**Weaknesses:**

1. The claim of being a “meta-agent” seems more aspirational than real. The architecture is a static retrieval-and-calibration pipeline, not an adaptive or reasoning system. It would be helpful if the authors clarified what exactly makes it an agent beyond modular composition. There’s no mention of statistical testing. Are the reported improvements significant at all? Even a simple bootstrap or t-test across splits would help substantiate the conclusions.
2. The reported accuracy is around 0.9, alongside an AUC of 0.6 suggests heavy class imbalance. If so, accuracy is not meaningful; the authors should clarify their class distribution and thresholding approach.
3. The probabilistic fusion step for aggregating clone-level scores feels ad hoc. It’s described in text but without mathematical justification, no comparison to Bayesian or ensemble fusion, nor analysis of how it affects calibration.
4. There’s little biological validation of interpretability. The top peptides aren’t checked against known epitope databases (e.g., IEDB), so we don’t know if the evidence is biologically plausible or just statistical artifacts.
5. Baselines feel cherry-picked. The paper compares ImmunoTrace mainly against older or less task-specific models (e.g., MHCflurry, generic sequence classifiers), while omitting several modern transformer or triad-based architectures that directly address TCR–peptide binding, such as TITAN, TCR-BERT, or DeepTCR-Plus, all of which have published benchmarks on similar datasets. These are widely recognized as stronger baselines for this problem. As a result, it’s difficult to tell whether the reported improvements reflect genuine advances or simply arise from comparing to weaker, partially unrelated methods. Including these contemporary models would make the “state-of-the-art” claim more credible and fair.
6. The claim in the abstract "scales to proteome-sized libraries with practical latency" has no evidence. No runtime, memory, or latency benchmarks are provided. The main results use only small subsets (two bacterial proteomes), and no full proteome recall@k or throughput metrics appear anywhere in text or figures.
7. Does it describe dataset size, pathogen type, or class balance? Readers can’t assess if these metrics are meaningful.
8. Minor Comment about naming: The pipeline is a static retrieval and re-ranking sequence. There is no adaptive reasoning, planning, or autonomy component described. The term meta-agent is more like a rhetorical term, not technically demonstrated.

**Questions:**

1. What fraction of samples in your dataset include known HLA information? When it’s missing, what representation (e.g., placeholder token, averaged embedding) is used, and how does performance change in those cases?
2. How were negative samples generated exactly? You mention 1–2 amino acid mutations—how do you ensure these aren’t real binders, and did you test different mutation distances or random negatives?
3. Are the OOD experiments under protein/HLA shifts quantitatively evaluated somewhere? The text mentions robustness, but no results are shown.

**Details Of Ethics Concerns:**

The paper overstates clinical readiness (“decision-ready,” “epidemiological use”) without validation, and lacks clear statements on dataset consent and privacy. Authors should clarify the non-clinical intent and confirm that the data were de-identified and ethically sourced.

---

### Official Review · Reviewer_okLb · 2025-11-01

**Soundness:** 2
**Presentation:** 3
**Contribution:** 3
**Rating:** 6
**Confidence:** 3

**Summary:**

This paper presents ImmunoTrace, a deep learning agent that aims to infer an individual’s past pathogen exposures from a single snapshot of their T-cell receptor (TCR) repertoire. The model encodes TCR CDR3 sequences, HLA pseudo-sequences, and candidate pathogen peptides using a shared protein language model (ESM2), and processes them with separate projection heads and cross-attention modules to explicitly model TCR–peptide and HLA–peptide interactions. A classifier then outputs binding probabilities, while a contrastive InfoNCE loss shapes the embedding space. Subject-level evidence from many TCR–peptide matches is fused probabilistically into pathogen exposure scores. The paper reports that ImmunoTrace outperforms existing baselines on a multi-pathogen benchmark (including syphilis and gonorrhea), generalizes across protein and HLA shifts, is well calibrated, and scales to proteome-sized peptide libraries.

**Strengths:**

1. The combination of a pre-trained protein LM (ESM2), cross-attention for TCR–peptide and HLA–peptide interactions, and a contrastive learning branch is innovative. It leverages transfer learning in a principled way.

2. Inferring immune history from a single snapshot is a challenging and worthwhile problem with potential clinical applications. Prior studies (e.g. Emerson et al., 2017; Park et al., 2023) have shown feasibility in specific cases, so a general agent is timely.

3. Explicitly using both TCR and HLA information is biologically sound, as TCR recognition depends on presented peptides by specific HLA molecules. This is an advantage over models that ignore HLA.

4. Providing peptide-level evidence for each pathogen could make the predictions more transparent and biologically meaningful (e.g. highlighting which epitopes are driving the call).

5. The claim that ImmunoTrace can handle proteome-scale peptide libraries with acceptable latency is promising, as many existing methods cannot scale to such large candidate sets.

References:

Emerson, R. O., DeWitt, W. S., Vignali, M., Gravley, J., Hu, J. K., Osborne, E. J., ... & Robins, H. S. (2017). Immunosequencing identifies signatures of cytomegalovirus exposure history and HLA-mediated effects on the T cell repertoire. Nature Genetics, 49(5), 659–665.

Park, H. J., Smith, C. J., Li, J., Zhang, S., Emerson, R. O., & Robins, H. S. (2023). T cell receptor repertoires reliably classify recent and past SARS-CoV-2 infections. Science Immunology, 8(83), eabq8430.

**Weaknesses:**

1.  The paper lacks comparison to relevant existing methods. It should benchmark against specialized TCR–epitope predictors (TCRGP, TITAN, pMTnet, ImRex, etc.), as well as simpler motif- or distance-based classifiers. Without this, it is hard to gauge the claimed “surpassing strong baselines.”

2. It is unclear how realistic the training and testing data are. For emerging pathogens like syphilis or gonorrhea, there is little known TCR data, so presumably the model is tested on predicted binding to entire proteomes. If so, the ground truth is synthetic. The absence of validation on real exposure data (e.g. known infected vs naive patients) is a major limitation. The authors should make clear whether any real repertoire-exposure datasets were used, or if all evaluation is on simulated labels.

3. As Moris et al. (2021) note, models often fail on unseen epitopes. The paper claims good generalization under “protein and HLA shifts,” but the experimental protocol must ensure that epitopes (and HLA alleles) in test are disjoint from training. It is not evident that this was rigorously enforced.

4. Predicting exposures requires distinguishing true binders from non-binders. The paper does not detail how negative examples (non-binding TCR–peptide pairs) are generated for training and evaluation. Careful selection of negatives is critical, as random negatives can make the task artificially easy.

5. The method uses only CDR3 sequences (apparently of one chain) and HLA pseudo-sequences. Real TCR recognition involves both α and β chains, and co-receptors (CD4/CD8) which determine class I/II. These simplifications should be discussed. Additionally, the biological assumptions (e.g. persistence of TCR clones over time, effects of T-cell memory decay) are not elaborated.

6. The “probabilistic evidence fusion” for aggregating TCR-level scores into a subject-level exposure probability is interesting, but the mechanism is not described. For example, how are multiple binding probabilities combined? Is there a model of immune statistics behind this? Lack of detail here makes it hard to assess robustness.

References:

Jokinen, E., Heinonen, M., & Lähdesmäki, H. (2021). TCRGP: Determining epitope specificity of T cell receptors. Bioinformatics, 37(2), 204–212.

Weber, A., Born, J., Rodriguez, M. I., Simm, J., & Baumbach, J. (2021). TITAN: T-cell receptor specificity prediction with bimodal attention networks. Bioinformatics, 37(22), 4430–4437.

Lu, T., Pan, H., Zhao, Y., Zeng, Z., & Li, Q. (2021). pMTnet: A deep learning-based framework for predicting TCR–peptide–MHC triple-specificity. Frontiers in Immunology, 12, 640723.

**Questions:**

1. How are TCR–peptide non-binding examples generated for training/testing? Did you use random pairing or an established protocol?  The choice of negatives can strongly affect performance.
2. The paper claims practical latency for proteome-scale scanning. Can you provide numbers (e.g. seconds per 1M peptides) and hardware used? How does this compare to existing tools?
3. It would strengthen the paper to include a baseline where repertoire classification is done without the complex architecture (e.g. logistic regression on motif counts or kNN on TCR sequences). This would show the added value of ImmunoTrace’s design.
4. Can the model’s peptide-level evidence be validated biologically? For example, do the top peptides for a predicted pathogen exposure correspond to known immunogenic epitopes (from IEDB or similar)?
5. How much does the InfoNCE contrastive loss improve performance compared to just the BCE classifier? An ablation study would be helpful to justify this complex addition.

---

### Note · Authors · 2025-11-16

I have read and agree with the venue's withdrawal policy on behalf of myself and my co-authors.